# [^18^F]MK-7246 for Positron Emission Tomography Imaging of the Beta-Cell Surface Marker GPR44

**DOI:** 10.3390/pharmaceutics15020499

**Published:** 2023-02-02

**Authors:** Pierre Cheung, Mohammad A. Amin, Bo Zhang, Francesco Lechi, Olle Korsgren, Jonas Eriksson, Luke R. Odell, Olof Eriksson

**Affiliations:** 1Science for Life Laboratory, Uppsala University, 751 83 Uppsala, Sweden; 2Department of Medicinal Chemistry, Uppsala University, 751 83 Uppsala, Sweden; 3Department of Immunology, Genetics and Pathology, Uppsala University, 751 83 Uppsala, Sweden

**Keywords:** ^18^F-labelling, beta-cell mass, diabetes, GPR44, CRTH2, islet imaging, PET

## Abstract

The progressive loss of beta-cell mass is a hallmark of diabetes and has been suggested as a complementary approach to studying the progression of diabetes in contrast to the beta-cell function. Non-invasive nuclear medicinal imaging techniques such as Positron Emission Tomography using radiation emitting tracers have thus been suggested as more viable methodologies to visualize and quantify the beta-cell mass with sufficient sensitivity. The transmembrane G protein-coupled receptor GPR44 has been identified as a biomarker for monitoring beta-cell mass. MK-7246 is a GPR44 antagonist that selectively binds to GPR44 with high affinity and good pharmacokinetic properties. Here, we present the synthesis of MK-7246, radiolabeled with the positron emitter fluorine-18 for preclinical evaluation using cell lines, mice, rats and human pancreatic cells. Here, we have described a synthesis and radiolabeling method for producing [^18^F]MK-7246 and its precursor compound. Preclinical assessments demonstrated the strong affinity and selectivity of [^18^F]MK-7246 towards GPR44. Additionally, [^18^F]MK-7246 exhibited excellent metabolic stability, a fast clearance profile from blood and tissues, qualifying it as a promising radioactive probe for GPR44-directed PET imaging.

## 1. Introduction

Diabetes is a disease affecting millions of people worldwide and is characterized by chronic hyperglycemia caused by defects in insulin production or insulin resistance in the peripheral tissues, leading to long-term complications including ulcers, retinopathy and neuropathy [1]. A common way to diagnose diabetes consists of measuring the function of beta cells through glucose metabolism indicators such as plasma glucose, glycated hemoglobin (HbA1c), insulin or its byproduct C-peptide. These indicators provide information on the beta-cell function (BCF) but poorly reflect the physical state of beta cells. More importantly, they correlate inaccurately with the disease progression, especially during the prediabetes phase as underlying complications due to chronic hyperglycemia could go undetected for many years [2]. Beta-cell mass (BCM), another important parameter of diabetes, has been suggested as a complementary focus point to evaluate disease progression due to its pathophysiological differences in type 1 (T1D) and type 2 (T2D) diabetes in contrast to the BCF [3,4]. However, contrary to the BCF, access to the BCM is restricted as it requires highly invasive and potentially harmful procedures such as pancreatic biopsies. The evolution of the BCM is thus still poorly understood prior to and following the onset of diabetes within human patients, marking a clear need for a precise assessment of alterations in the BCM to better understand the physiopathology behind the development of the disease. Lastly, the recovery of a functional BCM has been largely studied as a viable treatment approach for diabetes [5], pointing out the need for precise quantification and visualization of the BCM for designing new therapy strategies by measuring treatment effects and monitoring subsequent changes in the BCM.

In recent years, nuclear medicinal imaging techniques and particularly Positron Emission Tomography (PET) has been suggested as a potential tool for non-invasive and highly sensitive visualization and quantification of BCM [6]. Accordingly, the identification of a molecular target restricted to beta cells and available for molecular binding to a PET-compatible radioactive probe could greatly improve our understanding of diabetes etiology and enable new endpoints in therapy development [7]. The transmembrane G protein-coupled receptor GPR44 (also known as CRTH2 or PTGDR2) that binds to endogenous Prostaglandin D2 has recently been identified as a suitable biomarker candidate for monitoring BCM through proteomic and transcriptomic screenings [8]. Specifically, GPR44 is highly expressed in beta cells but not in other endocrine cells within the pancreatic islets, nor the surrounding exocrine tissue. Additionally, GPR44 expression was shown to be up to 40 times lower in the rat insulinoma cell line INS-1 [9]. Lower receptor availability means a lower proportion of bound radioactive tracer to the target biomarker, and thus, a reduced uptake signal within the tissue of interest. Therefore, the difference in receptor availability between the human and the rat pancreas further underlines the difficulties of the GPR44-targeted imaging of beta cells as it would require new models such as stem-cell-derived islets or large animal models (e.g., pigs).

The binding of GPR44 by endogenous prostaglandin D2 has been posited to inhibit insulin secretion via the activation of the Gαi/o subunit of GPR44 with a negative impact on cAMP production [10]. However, the oral administration of a GPR44 inhibitor showed no major impact on insulin secretion in patients with type 2 diabetes, but has demonstrated an improvement in islet function under inflammatory and hyperglycemic stress [11]. MK-7246 is a GPR44 antagonist with high affinity and good pharmacokinetic properties, originally developed to treat respiratory diseases [12]. Recently, we have developed [^11^C]MK-7246 as a potential PET tracer for the in vivo visualization of the BCM. [^11^C]MK-7246 displayed favorable biodistribution properties and in vivo specific binding to GPR44 within the pancreas [13,14]. Despite the attractive nature of carbon-11 as a radionuclide, the presence of a fluorine atom in the MK-7246 molecule warrants the development and evaluation of the fluorine-18-labeled isotopologue without influencing its kinetic properties. Fluorine-18 (t½ = 109.7 min, 0.633 MeV) can potentially provide a better imaging resolution due to the lower positron energy compared to carbon-11 (t½ = 20.4 min, 0.960 MeV) and superior clinical potential due to the longer half-life and simplified logistics. Herein, we describe the design and synthesis of [^18^F]MK-7246 and its preclinical evaluation as a novel in vivo marker for GPR44 and BCM. Notably, our novel synthetic strategy allows for access to both [^11^C]MK-7246 and [^18^F]MK-7246 from a common advanced intermediate, making the overall approach divergent and highly attractive.

## 2. Materials and Methods

### 2.1. Synthesis of BT4241 Precursor Compound for ^18^F-Labelling of MK7246

To a stirred solution of methyl 2-[7-[(4-nitrophenyl)sulfonylamino]-6,7,8,9-tetrahydropyrido [1,2-a]indol-10-yl]acetate (40.0 mg, 0.090 mmol, 1.0 equiv.) in dry DMF (1.0 mL), NaHMDS (1.0 M in THF) (24.8 mg, 0.135 mmol, 1.5 equiv.) was added in 135 μL of THF portion-wise at −40 °C under nitrogen atmosphere. After 30 min, a solution of iodomethane (6.7 μL, 0.108 mmol, 1.2 equiv.) in 0.5 mL of DMF was added dropwise and stirred for 3 h. The reaction mixture was quenched by pouring it into ice water and extracted with EtOAc (3 × 300 mL). The combined organic layer was washed with water and brine solution, dried with MgSO4 and concentrated in vacuo. The crude material was purified by 30% EtOAc in iso-hexane to afford the product as a yellow solid (25.0 mg, 61%). Rf = 0.35 (30% EtOAc in iso-hexane). 1H NMR (400 MHz, Chloroform-d) δ 8.49–8.34 (m, 2H), 8.10–8.02 (m, 2H), 7.58–7.49 (m, 1H), 7.21–7.09 (m, 3H), 4.58–4.45 (m, 1H), 4.24 (ddd, J = 11.2, 5.9, 1.2 Hz, 1H), 3.81 (t, J = 11.0 Hz, 1H), 3.73–3.55 (m, 5H), 3.15 (ddd, J = 16.6, 5.0, 2.8 Hz, 1H), 2.94 (s, 3H), 2.88 (td, J = 12.0, 6.2 Hz, 1H), 1.98–1.71 (m, 2H). 13C NMR (101 MHz, CDCl3) δ 172.3, 150.3, 145.4, 135.7, 132.1, 128.3, 128.2, 124.8, 121.4, 120.3, 118.3, 108.7, 103.1, 53.2, 52.1, 44.4, 30.0, 29.7, 25.2, 21.5. HRMS (ESI): Calc’d for C22H23FN3O6S [M + H]+ *m*/*z* 458.1386, found 458.1396. [α]25D = +105.04 (c 0.10, MeOH).

### 2.2. Radiochemistry

Tracer Production System (TPS, developed in house) was used for the automation of the radiosynthesis procedure. Semi-preparative high-performance liquid chromatography (prep-HPLC) was performed using an Agilent 1260 Infinity II pump, Agilent 1260 Infinity II UV detector, Bioscan Flow-count PMT radiodetector, TPS autosampler (developed in house) and a Gemini C18 5 µm 110 Å 250 × 10 mm column. The mobile phase used for the semi-preparative HPLC was 54% acetonitrile in ammonium formate (50 mM, pH 3.5) and ascorbic acid (0.2 M) at a flow rate of 5 mL/min. Analytical HPLC was performed using an Agilent 1290 Infinity II pump, Agilent 1290 Infinity II UV detector, Flow-count PMT radiodetector, and a Chromolith Performance RP-18e column (100 × 4.6 mm). The mobile phase was acetonitrile and 0.1% trifluoroacetic acid/water at a flow rate of 4 mL/min, and an acetonitrile gradient going from 5 to 73% over 8 min. 

All radiochemical yields (RCYs) were calculated based on the activity of the [^18^F]fluoride at the start of synthesis and the formulated [^18^F]MK-7246 product solution (decay corrected). The radiochemical purity (RP) and molar activity were assessed by analytical UV-radio-HPLC by analyzing a sample withdrawn from the formulated product solution at the end of synthesis (EOS). The identity of the labelled product was confirmed by spiking the sample with non-radioactive MK-7246 and comparing the retention time of the product peak on the UV and radio trace. 

Cyclotron-produced [^18^F]fluoride (approximately 10 µAh of irradiation) was passed through a QMA cartridge followed by nitrogen gas. Then, a solution (900 µL) of Kryptofix 2.2.2 (10 mg/mL) and potassium carbonate (1.4 mg/mL) followed by excess air was used to elute the [^18^F]fluoride from the QMA cartridge into the reaction vial. Drying of the resulting solution was performed at 120 °C for 8 min under a nitrogen flow (700 mL/min). Then, the precursor compound BT4241 (0.5–1.0 mg) in DMF (0.5 mL) was added and the mixture was heated at 150 °C for 10 min. The temperature was lowered to 90 °C and NaOH (0.6 mL, 1.0 M) was added and reacted at 90 °C for 5 min. After cooling to an ambient temperature, the reaction mixture was neutralized with HCl (0.6 mL, 1.0 M) and injected on the prep-HPLC system via an Alumina cartridge to remove unreacted [^18^F]fluoride (Sep-Pak Alumina N Plus Light Cartridge, Waters). The fraction eluting at 11.6 min was collected and diluted with water (20 mL) and passed through a solid phase extraction (SPE) cartridge (Sep-Pak C18 Plus Light, Waters) where the product was trapped. The SPE cartridge was then washed with water (20 mL) and eluted with ethanol (0.4 mL) followed by PBS with ascorbic acid (4 mL, 0.1 M) into a sterile product vial through a sterile filter (Millex-GV, 0.22 µm, 33 mm, Millipore).

### 2.3. Blood Metabolites Analysis

In vitro stability tests were performed with approximately 50 MBq of [^18^F]MK-7246 diluted with PBS to a total volume of 0.1 mL followed by incubation in approximately 1 mL of pig or human plasma. After 90 and 120 min, a sample was taken (0.3 mL) and protein precipitation was performed using an equal part of acetonitrile (0.3 mL), and the resulting mixture was centrifuged at 16,000× *g* for 1 min (Eppendorf 5415R centrifuge) at 4 °C. The supernatant was filtered (Acrodisc 0.2 µm Supor membrane low protein binding filter). Analytical HPLC was performed using a Hitachi Chromaster HPLC system (VWR), FC-3300PMT radiodetector, 5420 UV-VIS detector and 5110 pump with a RP-column Vydac 214MS C5 (50 × 4.6 mm). The mobile phase used for the analysis was 10% acetonitrile in 0.1% TFA (MilliQ water) at a flow rate of 1 mL/min with an acetonitrile gradient going from 10 to 100% over 10 min.

In vivo stability tests were performed by the intravenous administration of 5 MBq [^18^F]MK-7246 through the tail vein of the rat, and a blood sample of 1 mL was taken after 90 min. The blood sample was centrifuged at 3000× *g* for 2 min at 4 °C (Beckham Allegra X-22R centrifuge) and the plasma supernatant was collected and further processed as described above for the in vitro stability testing.

### 2.4. Cell Line Culture

The human GPR44-expressing Chinese hamster ovarian (CHO-K1) stable cell line was purchased from PerkinElmer and was cultured, according to the provider’s recommendations, in Ham’s F-12 (Gibco) supplemented with 10% FBS (Gibco), 0.4 mg/mL G418 (Corning), 1% Penicillin/Streptomycin (PEST) (Gibco) and 2 nM L-Glutamine (Gibco). Non-transfected CHO-K1 cells used as negative control were purchased from ATCC and cultured in Ham’s F-12 (Gibco) supplemented with 10% FBS (Gibco) and 1% PEST (Gibco). 

The human beta-cell line EndoC-BH1 was obtained from Endocells and cultured according to the provider’s indications in DMEM low glucose supplemented with 2% Bovine Serum Albumin (BSA) (Roche Diagnostics), 50 µM 2-mercaptoethanol (Sigma-Aldrich), 10 mM nicotinamide (VWR Life Science), 5.5 µg/mL transferrin and 6.7 ng/mL sodium selenite.

### 2.5. Human Tissue

The study has been conducted in accordance with the Declaration of Helsinki as well as the local and national guidelines. Anonymized human blood samples were obtained from the Uppsala University Hospital blood bank. Human islets were obtained from non-diabetic deceased donors within the Nordic Network for Clinical Islet Transplantation Laboratory (Uppsala University Hospital, Uppsala Sweden) through isolation methods described by Goto et al. [15]. 

All organ donors in Sweden provided written consent that their donated tissues may be entered into a biobank and used in medical research, following review and approval by the Regional Ethics Board (now Swedish Ethical Review Authority) of individual studies. The obtained tissues were anonymized, collected and treated according to local institutional and Swedish national rules and regulations, with the need for informed consent renounced by the Regional Ethics Board in Uppsala.

The use of human tissues from Uppsala Biobank (registration #827) was approved by the Regional Ethics Board, Uppsala, Sweden (2011/473, Ups 02-577, 2015/401).

### 2.6. Saturation Binding Curve and Specific Binding

GPR44-overexpressing CHO-K1 cells and non-transfected CHO-K1 cells were first detached using Trypsin 0.25% EDTA (Gibco) and gently washed with PBS. Washed cells were collected and suspended in polystyrene round-bottom tubes using PBS at room temperature (RT) prior to the start of the experiment. Cell count and viability were assessed using a TC-20 automated cell counter (Bio-Rad) using trypan blue (Bio-Rad) at a 1:1 ratio. To assess the specific binding, the suspended GPR44-expressing CHO-K1 or non-transfected CHO-K1 cells were incubated for 60 min at RT with 3 MBq/mL of [^18^F]MK-7246 (corresponding to approximately 15 nM) supplemented with either PBS, Fevipiprant (1 mM), AZD1981 (1 mM) or MK-7246 (1 mM) before suctioning the incubated cells through a Whatman filter (Milipore) using a Harvester for Liquid Scintillation Counters (Brandel). 

For the saturation binding curve, a concentration gradient ranging from 1.5 kBq/mL to 3 MBq/mL (equivalent to a concentration range of to 5 pM to 10 nM) of [^18^F]MK-7246 was used to incubate the GPR44-expressing CHO-K1 cells for 60 min at RT before suctioning the incubated cells through a Whatman filter (Milipore) using a Harvester for Liquid Scintillation Counters (Brandel). Both for the specific binding assessment and the saturation binding curve, patches of filters where the cells went through were collected and measured for radioactivity and decay, and were corrected using a well counter (Uppsala Imanet AB, Uppsala, Sweden).

### 2.7. In Vitro Autoradiography of Cell Pellets

GPR44-expressing CHO-K1 cells and non-transfected CHO-K1 cells were first detached using Trypsin 0.25% EDTA (Gibco) and gently washed with PBS. Washed cells were centrifuged for 5 min at 300× *g* to form a pellet. Pellets of human pancreatic exocrine fractions and endocrine islet fractions of varying purity (98%, 76%, 27%) were obtained through sedimentation. Each obtained pellet was embedded into an optimal cutting temperature (OCT) compound (Q Path mounting media, VWR) and frozen at −80 °C. Slices of 20 µm thickness were prepared on cryotome (Cryostat NX70, ThermoFisher) before mounting on SuperFrost Plus slides (ThermoFisher) and stored at −20 °C prior to the experiment. Frozen sections of CHO-K1 cells expressing GPR44, non-transfected CHO-K1 cells, exocrine pancreatic fractions and endocrine pancreatic islet fractions were incubated at RT for 60 min with PBS and 0.5 MBq/mL of [^18^F]MK-7246 (corresponding to approximately 2 nM). A series of two 3 min cold washes in PBS and one 1 min wash in deionized water was performed after the incubation. Samples were dried and left for exposure on the BAS-IP storage phosphor screen (Fujifilm) for 90 min. A total of 20 µL droplets of reference radioactive solution on absorbent paper, cross-measured in a well counter (Uppsala Imanet AB, Uppsala, Sweden) was also included to enable quantification of the autoradiograms. The resulting digital image readout was obtained using an Amersham Typhoon storage phosphor imager (GE healthcare).

Autoradiograms were analyzed using ImageJ (National Institute of Health) with the cell sections and references delineated manually. The uptake values were corrected for background counts and converted to Bq through a reference and reported as a ratio of added dose (Bq/MBqadded). 

### 2.8. Animal Handling

All the animal experimentations and handlings were approved by the local animal research ethical committee of the Uppsala region and performed according to the Uppsala University guidelines for animal research (16167/2019) and in respect of the ARRIVE guidelines. For each individual experiment involving either rats or mice, the anesthesia was given through the inhalation of 4% vaporized sevoflurane mixed with a 400 mL/min flow of medical air and oxygen.

### 2.9. Dynamic PET/MRI and Biodistribution in Rats

A total of n = 8 rats (male, Sprague Dawley, weight of 250–300 g) were ordered and transported to the animal facility a week prior to each individual experiment for acclimatization. Rat PET imaging was performed to identify the optimal imaging time point for future studies in other in vivo models. Furthermore, the rat model was used to establish in vivo stability and dosimetry of the tracer. As opposed to humans, the beta cells from rats do not express high levels of GPR44 and strong pancreas binding was not expected.

A magnetic resonance imaging (MRI) acquisition for attenuation and localization was first performed on n = 2 rats (350 ms TR, 3.9 ms TE, 146 slices, 13 Th, 0 Gap, 80 mm FOV, 0.5 mm resolution, 2 NEX, 10 min acquisition time). Subsequently, around 15 kBq/g (total amount of 5 MBq) of [^18^F]MK-7246 dissolved in PBS and 10% ethanol were injected intravenously through the tail vein of each rat. A dynamic PET imaging of 150 min was then acquired using the following parameters: 8 frames of 2 × 1.4′, 2 × 3.2′, 4 × 10′, 0.4 × 0.4 × 0.4 mm voxels, 212 × 212 × 579 matrix size. All the sequences described previously were performed with a nano PET/MRI scanner (Mediso). PET data were analyzed with manual segmentation on sequential transaxial projections using the PBAS modeling tool (PMOD technologies LLC). The uptake in kBq/cc was converted to Standardized Uptake Values (SUV) by correcting for the administered radioactivity and the weight of each rat. The radioactive dosimetry was also extrapolated from the rat data to human, based on reference measurements obtained from both the International Commission on Radiological Protection publication 89 (ICRP89) annals [16] and from the Medical Internal Radiation Dose (MIRD) model values summarized by Stabin and Siegel (2003) [17]. The residence time (RT) was calculated from the area under the curve (AUC) of the time–activity curve. The AUC was calculated using the trapezoid method for the time points 5, 10, 20, 30, 60, 90, 120, 150 min post-injection in addition to the tail area, calculated by assuming a monoexponential decay starting from the 150 min time point. The absorbed dose to each individual organ and the whole-body effective dose (ED) were obtained by OLINDA/EXM 1.1 (Organ Level Internal Dose Assessment Code, Vanderbilt University, Nashville, TN).

For the endpoint biodistribution, an injection of 15 kBq/g [^18^F]MK-7246 (total amount of 5 MBq) through the lateral tail vein of n = 3 rats was performed for baseline measurements, whereas 50 µL of GPR44 antagonist Fevipiprant (20 mM) formulated in PBS was injected intravenously by bolus 30 min prior to injection of 15 kBq/g [^18^F]MK-7246 (total amount of 5 MBq) through the lateral tail vein of n = 3 rats. Euthanasia of the rats was conducted 90 min post-injection and organs were collected for measurement in the well counter (Uppsala Imanet AB, Uppsala, Sweden).

### 2.10. Dynamic PET/CT and Biodistribution in Mice

A total of n = 14 female BALBc nu/nu mice weighing 20–30 g were ordered and transported to the animal facility. Subcutaneous injections into the left hind leg of each mouse with either GPR44-overexpressing CHO-K1 cells or non-transfected CHO-K1 cells were performed. Tumors were allowed to develop for roughly 1 month to a tumor size of maximally 1 cm in diameter. 

A computed tomography (CT) acquisition for attenuation and localization was first performed (50 kVp, 610 µA, 480 proj, 1:4 binning, 251 × 251 × 251 µm voxel size) before injecting [^18^F]MK-7246 (250 kBq/g, total amount of 5 MBq) in n = 3 mice with either a non-transfected CHO-K1-cell-induced tumor, a GPR44-expressing-CHO-K1-induced tumor or a GPR44-overexpressing-CHO-K1-induced tumor supplemented with Fevipiprant (80µL, 20 mM) through the lateral vein 30 min prior to injection of [^18^F]MK-7246. A static PET imaging of 30 min was then acquired (1 × 30′ frame, 0.4 × 0.4 × 0.4 mm voxels, 212 × 212y 235z matrix size) 60 min post-injection. All the sequences described previously were performed on a nanoPET/MRI and a nanoSPECT/CT scanner (Mediso).

For the endpoint biodistribution, an injection of 50 kBq/g [^18^F]MK-7246 (total amount of 1 MBq) was administered through the lateral tail vein of the mice (n = 11) with either a non-transfected CHO-K1 cell-induced tumor or GPR44-expressing-CHO-K1-induced tumor. Euthanasia of the mice was conducted 60 min post-injection and organs were collected for measurement in the well counter (Uppsala Imanet AB, Uppsala, Sweden).

### 2.11. Statistics

All statistical analyses were performed on Graphpad prism (version 8.3.1). Differences between groups were assessed by either one-way ANOVA with Bonferroni’s correction for multiple comparisons, student *t*-test or Welch’s *t* test using *p* < 0.05 as the limit for significance.

## 3. Results

### 3.1. Precursor Synthesis

Synthesis of the target 4-nitrosulfonamide containing precursor BT4241 was achieved in 14 liner steps in an overall yield of 0.4%. The desired stereochemistry was installed using a chiral pool approach starting from D-aspartic acid and extensive elaboration led to the primary amine intermediate BT4235 (Figure 1). Introduction of the key ^18^F-precursor aryl-nitro motif was achieved by treatment with 4-nitrobenzenesulfonyl chloride, and finally, methylation in the presence of NaHMDS afforded the target precursor BT4241 in 61% yield. 

### 3.2. Synthesis of [^18^F]MK-7246

The radioactivity yield of [^18^F]MK-7246 was 340 ± 7 MBq and the RCY was 3 ± 0% (entries 1–3) when using 0.5 mg of precursor and was increased to 583 ± 106 MBq (entries 4–6) and 6 ± 1%, respectively, when using 1.0 mg of precursor (Table 1). The radiochemical purity was initially 87% (entry 1) and was increased to >97% (entries 2–6) with the addition of ascorbic acid in the aqueous preparative HPLC eluent. The molar activity was approximately 400 GBq/µmol (Figure 2).

### 3.3. Blood Metabolites Analysis

The in vitro blood metabolite analysis of [^18^F]MK-7246 performed in both pig and human plasma showed a >95% intact tracer after 120 min incubation. The in vivo stability in rat was also high with a >95% intact tracer in plasma 90 min after injection (Appendix A). 

### 3.4. Saturation Binding Curve and Binding Specificity

Using the one-site-specific-binding-nonlinear-fit function of Graphpad Prism, the maximum receptor availability (Bmax = 2.07 MBq/mL (95% CI: 1.88 to 2.31 MBq/mL)) and the affinity (Kd = 1.61 (95% CI: 1.31 to 2 MBq/mL) were calculated. Specific activity related to the tracer production ranged from 200 to 600 MBq/nmol, thus, the affinity is expected to be in the nanomolar range of 2.68–8.05 nM (Figure 3).

The binding signal of [^18^F]MK-7246 to GPR44-expressing CHO-K1 was significantly higher compared with the non-transfected CHO-K1 negative control. More importantly, the binding signal could be suppressed in the GPR44-expressing CHO-K1 cells by saturating the GPR44 receptors using GPR44 antagonists of various molecular classes such as Fevipiprant, AZD1981 or non-radioactive MK-7246 (**** *p* < 0.0001) (Figure 4). [^18^F]MK-7246, thus, demonstrated excellent selectivity toward GPR44 receptors in addition to minimal non-specific binding.

### 3.5. In Vitro Autoradiography

Frozen sections of CHO-K1 cells expressing GPR44 (uptake signal = 664 ± 298.5 Bq/MBqinj, n = 10) showed a significantly higher binding of [^18^F]MK-7246 compared with frozen sections of non-transfected CHO-K1 negative controls (uptake signal = 34.73 ± 7.89) (**** *p* < 0.0001) (Figure 5A).

Furthermore, frozen sections of EndoC-BH1 cells (uptake signal = 139.7 ± 29.74 Bq/MBqinj, n = 11) and the 98% purity endocrine fraction (uptake signal = 72.45 ± 13.54 Bq/MBqinj, n = 13) showed a significantly higher binding of [^18^F]MK-7246 (**** *p* < 0.0001 and ** *p* < 0.001) compared with frozen sections of the exocrine fraction (uptake signal = 54.91 ± 10.85 Bq/MBqinj, n = 46). Neither 76% (uptake signal = 60.95 ± 15.65 Bq/MBqinj, n = 22) nor 27% endocrine islets (uptake signal = 48.77 ± 12.5. Bq/MBqinj, n = 6) showed a significant binding of [^18^F]MK-7246 compared to the exocrine pancreas tissue (Figure 5B). 

### 3.6. In Vivo Biodistribution and Dosimetry in Rats

The dynamic biodistribution of [^18^F]MK-7246 in rats (n = 2) over the 150 min PET/MRI demonstrated low signal uptake in the majority of the organs, except excretion organs such as the liver. As opposed to humans, beta cells of rodents do not express high levels of GPR44, and thus, the pancreas was not visible. The kidney displayed a fast uptake of [^18^F]MK-7246 followed by a rapid washout, whereas the liver displayed a high uptake of [^18^F]MK-7246 followed by excretion starting at 60 min. The liver excretion corroborated the simultaneous signal increase of [^18^F]MK-7246 in the small intestines where the bile containing the radioactive compound was discharged (Figure 6). 

The endpoint biodistribution from the collected organs aligned with the dynamic PET/MRI results with [^18^F]MK-7246 metabolized by the liver. No differences in the biodistribution profiles could be observed between baseline rats and rats injected with the GPR44 antagonist Fevipiprant (Figure 7). As mentioned, the pancreatic islets of rodents do not express GPR44 to the same extent as humans or large animals do. Thus, the absence of [^18^F]MK-7246 binding the pancreas was expected to a certain extent. 

The absorbed radioactive dose was highest for the liver where the main excretion of [^18^F]MK-7246 occurs (Figure 8). The predicted human effective dose (ED) extrapolated from the RT in rats is 14.55 ± 2.75 µSv per MBq of injected [^18^F]MK-7246.

### 3.7. In Vivo Biodistribution of Subcutaneous Grafts in Mice

The dynamic PET/CT of the mouse with a tumor developed from GPR44-expressing CHO-K1 cells showed a clear uptake signal of [^18^F]MK-7246 (SUV = 1.37) compared to the mouse with a tumor developed from non-transfected CHO-K1 cells (SUV = 0.25). More importantly, the administration of Fevipiprant prior to the injection of [^18^F]MK-7246 could decrease the uptake signal in the mouse with a tumor developed from GPR44-expressing CHO-K1 cells (SUV = 0.59) (Figure 9A). The endpoint biodistribution of [^18^F]MK-7246 in BALB/C nu/nu mice with a tumor developed from CHO-K1 cells expressing GPR44 displayed a significantly higher radioactive signal uptake in the tumor area compared to tumors developed from the implantation of non-transfected CHO-K1 (**** *p* < 0.0001) (Figure 9B). The biodistribution showed a high uptake of [^18^F]MK-7246 in the liver that could be attributed to excretion, but no significant differences could be observed in the biodistribution profile of other organs between mice implanted with CHO-K1 cells overexpressing GPR44 or non-transfected CHO-K1.

## 4. Discussion

PET imaging of the pancreatic BCM faces many challenges, including the heterogeneous distribution of islets across the pancreas and its non-uniform microarchitecture. Furthermore, the intrinsic resolution of PET imaging imposed by the clinical PET scanner (~3–5 mm) prevents the identification of individual islets due to their small size (~20–500 µm). Therefore, the signal from the BCM would be represented as the sum of all pancreatic beta cells proportional to the whole imaged pancreas [18] using a similarly perfused organ as reference, such as the spleen. Those restrictions thus require the development of a highly selective and sensitive PET imaging probe, matched with a highly expressed target and restricted to the BCM. Ideally, the generated probe should display a signal many folds higher in the endocrine islets compared to the exocrine tissues, but also sufficiently higher than the non-specific signal arising from the surrounding tissues as well as the plasmatic unbound tracer associated with organ perfusion.

Given our continued interest in clinical translation, our synthetic strategy focused on the development of a modular pathway that would allow for the synthesis of either [^11^C]MK-7246 previously described by Eriksson et al. and Cheung et al. [13,14] and its isotopologue [^18^F]MK-7246 from a common advanced intermediate. In our previous work, the fluorosulfonamide group was introduced in the third synthetic step and this approach was deemed unsuitable for further development. In light of the available literature on MK-7246 [19], we identified the primary amine containing intermediate BT4235 as an excellent candidate for late-stage diversification into both isotopologues by simply exchanging the sulfonyl-chloride reactant in the final synthetic steps. The methyl-ester-protecting group was retained as optimized conditions for its removal had already been developed. Synthesis of compound BT4235 was achieved in 12 steps starting from D-aspartic acid and methyl indole-3-acetate. Treatment of BT4235 with 4-fluorobenzenesulfonyl chloride afforded the [^11^C]MK-7246 precursor BT4236 with 88% yield, whereas its reaction with 4-nitrobenzenesulfonyl chloride and methylation led to target the [^18^F]MK-7246 precursor BT4241 in 38% yield over two steps (Figure 10). 

The synthesis of [^18^F]MK-7246 was first attempted by eluting the [^18^F]fluoride from the QMA with tetraethylammonium bicarbonate and circumventing the prior drying step reaction with the labelling precursor, however, no product was formed. We then shifted to the conventional ^18^F-fluorination conditions using potassium kryptofix and potassium carbonate and obtained small amounts of product when performing the reaction at 150 ºC for 10 min. After changing the solvent from DMSO to DMF, the activity yield significantly improved from 45 MBq to approximately 340 MBq. Further adjustment of the reaction temperature and time resulted in lower radioactivity yields. As expected, doubling the amount of precursor led to an increase in the RCY with a radioactivity yield averaging 580 MBq. Interestingly, the deprotection conditions developed for the [^11^C]MK-7246 were unsuccessful when applied to the reaction mixture obtained from fluorine-18 labelling. Complete deprotection was achieved by increasing the temperature from 75 °C to 90 °C and the reaction time from 1 min to 5 min. The HPLC column was also changed as the previously used ACE C18 column promoted on-column radiolysis. However, radiolysis was still observed in the formulated product solution, and this was reduced by the addition of ascorbic acid to the fraction collection vial and the formulation solution. 

The affinity of [^18^F]MK-7246 towards GPR44 was assessed in vitro and was found to be in the nanomolar range, in line with previously reported values for non-labeled MK-7246 [20]. Thus, the substitution of a fluorine atom to fluorine-18 did not affect the function and binding capacity of MK-7246 as expected. Moreover, we also demonstrated the binding specificity of [^18^F]MK-7246 to GPR44 through an in vitro cell line model which confirmed that [^18^F]MK-7246 retained the specificity compared with the previously described [^11^C]-MK7246. Finally, the plasma stability of [^18^F]MK-7246 was high in various animal species, and proof of concept in vivo imaging was demonstrated in GPR44-expressing tumors. 

Biodistribution in rats and mice exhibited excretion mainly through the hepatic-biliary-intestinal axis with low transit through the kidney, which is expected to a certain extent from a small molecule compound with lipophilic properties. Pancreatic uptake of the tracer was low in rats (SUV < 1) and similar to a negative control tissue such as the muscle. This could be explained by the difference in islet microarchitecture in rodents (i.e., made up of a mantle of alpha cells with beta cells at the center) [21]. Moreover, beta cells of rodents only express low levels of GPR44 in comparison to larger animals such as pigs, non-human primates or humans, as shown in previous studies using the rodent beta-cell line INS-1 by Hellström-Lindahl et al. [9], reiterated in the present study as no significant differences in the pancreatic uptake signal of [^18^F]MK-7246 could be observed in the presence or absence of the GPR44 antagonist Fevipiprant.

The optimal biodistribution window for [^18^F]MK-7246 was determined to be later than 60 min post-injection when the hepato-biliary elimination of the radiopharmaceutical had started. The sharp decrease in [^18^F]MK-7246 uptake signal in the liver aligned with the simultaneous increase in the tracer uptake signal in the small intestine, reflecting the excretion process of the radiopharmaceutical compound through the bile. Assuming similarities between [^18^F]MK-7246 and its previously evaluated carbon-11 counterpart, a clearly defined pancreas with no spillover to adjacent tissues is predicted in a large animal model but with an improved resolution due to the lower energy range of fluorine-18. [^18^F]MK-7246 also demonstrated a safe human dosimetry extrapolated from rats. Assuming a clinical dosing of 200 MBq [^18^F]MK-7246 (approximately 2.85 MBq/kg for an adult of 70 kg) matching a whole-body effective dose of 1.46 mSv and the addition of a necessary CT scan over the abdomen (~1 mSv) for attenuation correction and anatomical co-registration, up to 3 PET/CT examinations per year would still be acceptable before reaching the 10 mSv yearly limit set for clinical research studies on young adults in Sweden. The lack of GPR44 expression within the rat pancreas would most definitely cause an underestimation of the pancreatic dose, but should not significantly influence the dosimetry of the other tissues as the main factors influencing the dosimetry come from excretion and biodistribution.

The in vitro frozen section autoradiography showed a strong binding of [^18^F]MK-7246 to the GPR44 expressed CHO-K1 cells but not in the negative-control CHO-K1 cells. More importantly, the engineered human pancreatic beta-cell line EndoC-BH1 and the highly pure pancreatic endocrine fraction (98%) showed a significantly higher uptake signal of [^18^F]MK-7246 compared to the exocrine fraction. The weak average density of available GPR44 receptors on the beta cells—as displayed by the lack of significant binding signal to endocrine fractions with lower purity, coupled with a low proportion of pancreatic islets (~1–2%) within the pancreas—could be critical shortcomings for a potential BCM PET marker. However, given the signal represents highly specific binding of the surface receptor GPR44, the overall signal derived from the total pancreas would yield a good quantitative estimate of the target density. Additionally, the lack of binding to lower purity islets should not be a limiting factor as beta cells in the pancreatic volume are naturally dispersed. The imaging of each single pancreatic islet would be impossible, but the signal from the BCM would be represented as the sum of all pancreatic beta cells proportional to the whole imaged pancreas using a similarly perfused reference organ such as the spleen. Therefore, the bottleneck would be to obtain a probe displaying a signal many folds higher in endocrine islets compared to exocrine tissues, but also significantly higher than the non-specific signal arising from the surrounding tissues as well as the unbound plasmatic tracer associated with organ perfusion.

As the density of available receptors is low, the importance of the specific activity (i.e., the ratio of the radioactive tracer in regard to its non-radioactive part) increases as to avoid risks of saturation of the receptor population by the non-radioactive MK-7246. However, given the low doses expected for administration, we predict a negligible occupancy of GPR44 in the pancreas, and thus, no measurable pharmacological effect. Consequently, [^18^F]MK-7246 at a microdose level is likely to be a safe compound for repeated PET imaging in humans.

[^18^F]MK-7246 presents several potential advantages for in vivo PET imaging of GPR44 compared to a [^11^C]MK-7246 isotopologue. Both analogues demonstrated optimal imaging contrast 60 min after administration, but at that time point, [^18^F]MK-7246 will have more remaining counts in the tissue, whereas three half-lives would have already passed for [^11^C]MK-7246. Counts above background will likely be especially important for targets with low density in tissue, e.g., dispersed beta cells in the pancreas. Additionally, the Fluorine-18 nuclide has intrinsically higher spatial resolution compared to Carbon-11 due to lower energy, and hence, shorter positron range. This may be a crucial advantage for the pancreas, which is an oblong organ with a diameter of only a few centimeters. Given that the resolution of modern clinical PET scanners is in the range of 4–5 mm, optimal resolution will minimize spillover and partial volume effects that could influence quantification. However, [^18^F]MK-7246 clearly has a more challenging dosimetry profile than [^11^C]MK-7246 due to its longer radioactive half-life, which may be an important factor when considering PET imaging in vulnerable populations such as young adults or even children. On the other hand, this same property will enable facile logistics and the distribution of [^18^F]MK-7246 between PET sites, potentially increasing availability.

The close anatomical proximity of the liver and intestines with the pancreas could potentially pose a spill-in issue, thus, highly impairs the possibility of native beta-cell imaging. However, the distance between the pancreas and the liver or kidneys will be significantly larger in humans. Therefore, only parts in very close proximity may be subject to spill-in. The issue is mostly prevalent in smaller animals such as rodents (<300 g) and non-human primates (5 kg), but much less significant in larger animals such as pigs (30 kg) or humans (80 kg). Future studies with a GMP-compliant production method of MK-7246 in addition to clinical evaluations would further assess those risks. The use of [^18^F]MK-7246 for insulinoma PET imaging could be potentially considered, but no data are currently available regarding the GPR44 expression on insulinoma cells [22].

## 5. Conclusions

We have described a robust synthesis and radiolabeling method for producing [^18^F]MK-7246, an antagonist to GPR44, a receptor protein expressed specifically on the beta cells within the pancreatic islets. Subsequent preclinical in vitro and in vivo assessments underlined the strong affinity and selectivity of [^18^F]MK-7246 towards GPR44. Additionally, [^18^F]MK-7246 exhibited excellent metabolic stability with a fast clearance profile from blood and tissues, thus offering a low non-specific signal, better contrast and an overall safe dosimetry profile suitable for repeated PET scans. The results from this study support the further development of [^18^F]MK-7246 for the in vivo PET imaging of BCM.

## Figures and Tables

**Figure 1 pharmaceutics-15-00499-f001:**
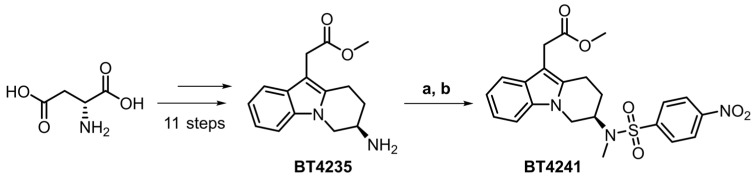
Chiral pool approach to access key target precursor BT4241 starting from D-aspartic acid. (**a**) 4-nitrobenzenesulfonyl chloride (2.0 equiv.), Et3N (3.1 equiv.), THF, rt, 12 h, 62%. (**b**) MeI (1.2 equiv.), NaHMDS (1.5 equiv.), DMF/THF, −40 °C, 3 h, 61%.

**Figure 2 pharmaceutics-15-00499-f002:**
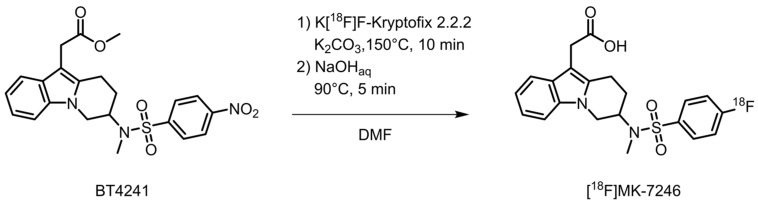
Synthesis of [^18^F]MK-7246.

**Figure 3 pharmaceutics-15-00499-f003:**
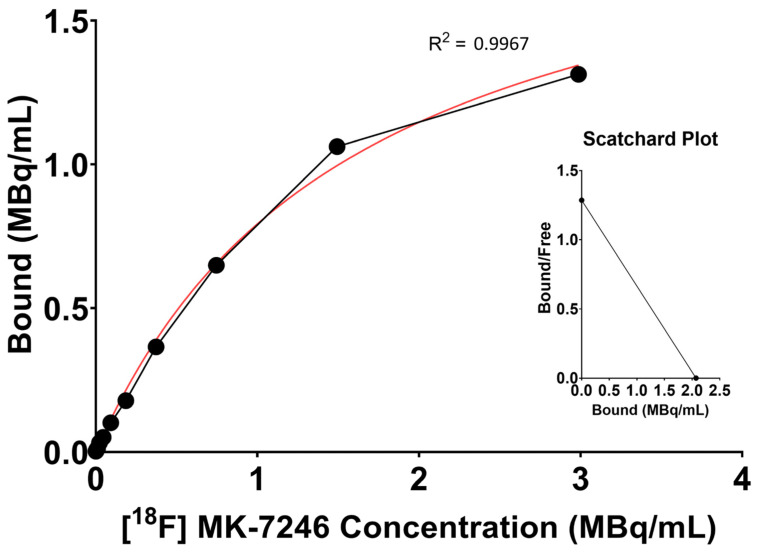
Representative saturation binding curve and Scatchard plot calculated from the binding of [^18^F]MK-7246 to GPR44-expressing cells using the one-site-specific-binding-nonlinear-fit function of Graphpad Prism (average value of n = 15).

**Figure 4 pharmaceutics-15-00499-f004:**
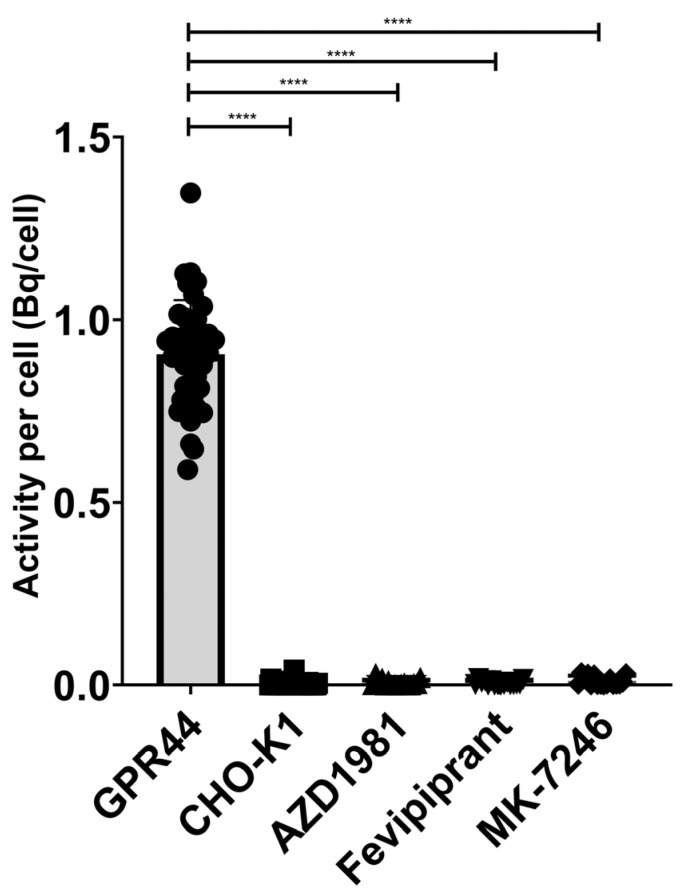
Binding signal of [^18^F]MK-7246 to GPR44-expressing cells (n = 42) were significantly higher compared to the non-transfected CHO-K1 cells (n = 36) (**** *p* < 0.0001). [^18^F]MK-7246 uptake signal in GPR44-expressing cells was decreased by saturating the GPR44 receptors with AZD1981 (n = 16), Fevipiprant (n = 16) or non-radioactive MK-7246 (n = 16).

**Figure 5 pharmaceutics-15-00499-f005:**
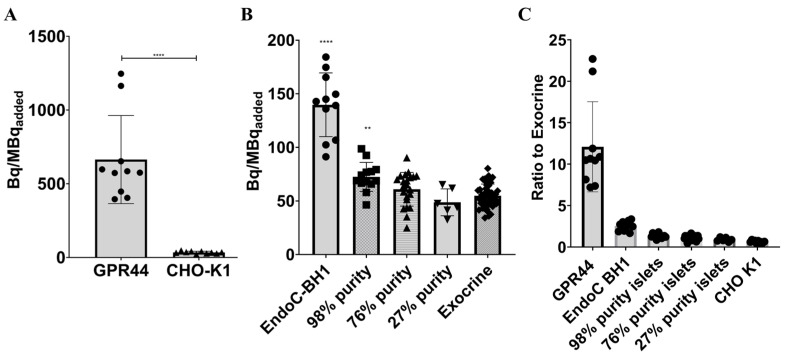
(**A**) Binding signal of [^18^F]-MK7246 to CHO-K1 cells expressing GPR44 (n = 10) were significantly higher compared with non-transfected CHO-K1 cells (n = 10) (**** *p* < 0.0001), (**B**) Binding signal of [^18^F]-MK7246 to EndoC-BH1 (n = 11) and human pancreatic fractions of various purity 98% purity (n = 13) were significantly higher than the exocrine fraction (n = 46) (**** *p* < 0.0001 and ** *p* < 0.001 respectively). Neither the 76% (n = 22) or the 27% (n = 6) showed any significant binding of [^18^F]MK-7246 compared to the exocrine pancreas tissue). (**C**) Uptake ratio to human exocrine fraction of [^18^F]-MK7246.

**Figure 6 pharmaceutics-15-00499-f006:**
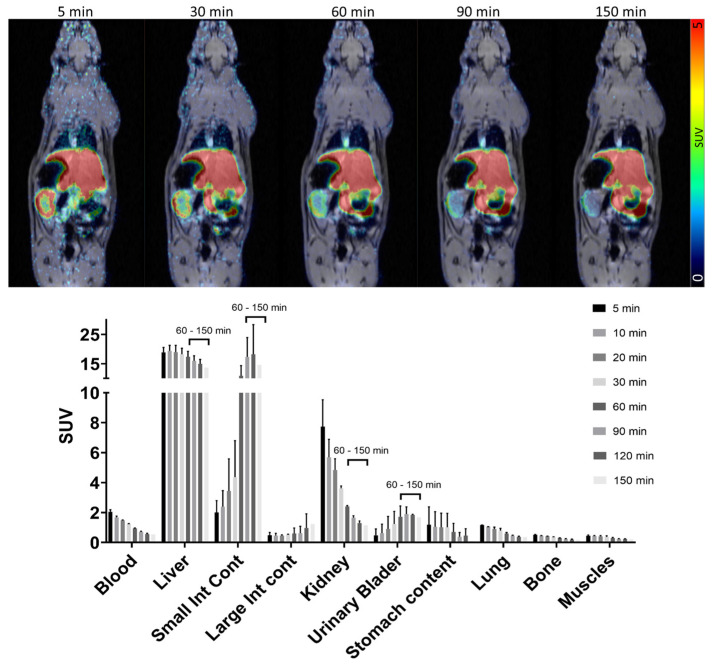
PET/MRI dynamic biodistribution of [^18^F]MK-7246 in n = 2 rats over 150 min.

**Figure 7 pharmaceutics-15-00499-f007:**
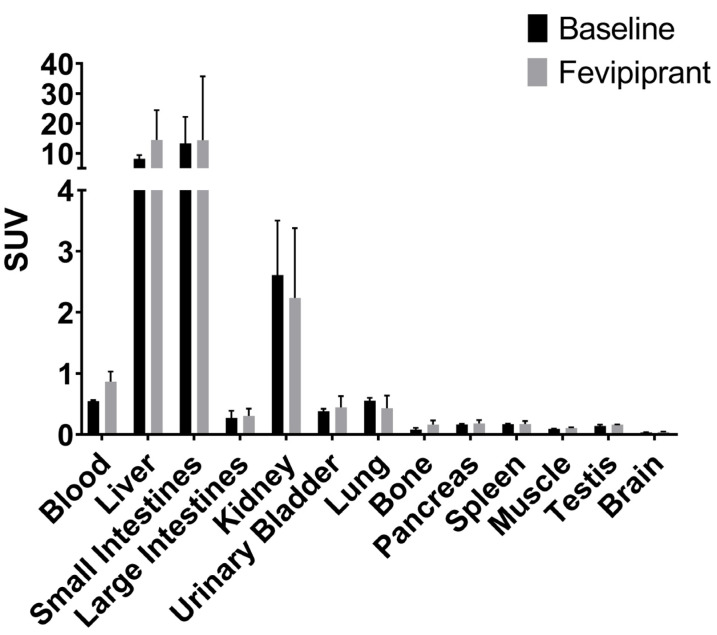
Endpoint biodistribution 90 min post-injection of [^18^F]MK-7246 in n = 3 baseline rats and n = 3 rats injected with GPR44 antagonist Fevipiprant (1 µmol).

**Figure 8 pharmaceutics-15-00499-f008:**
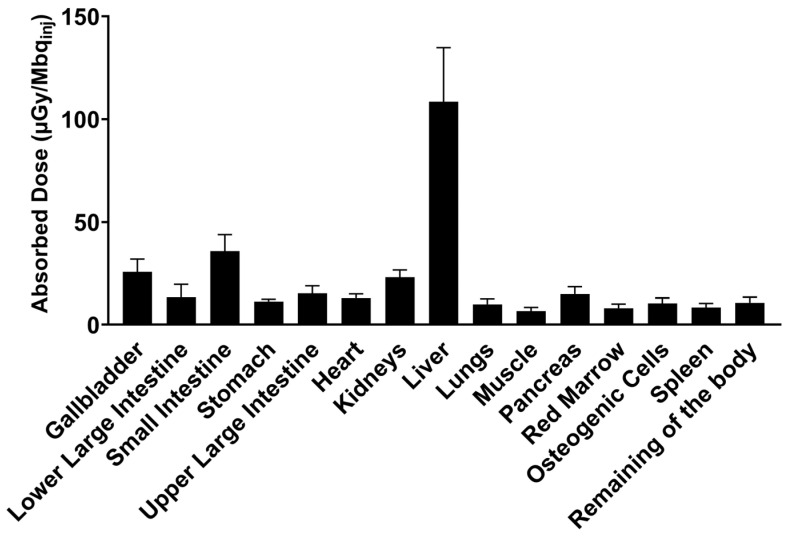
Human predicted dosimetry of [^18^F]MK-7246 extrapolated from n = 2 rat biodistribution, expressed as the absorbed dose per unit of administered radioactivity (µGy/Mbqinj).

**Figure 9 pharmaceutics-15-00499-f009:**
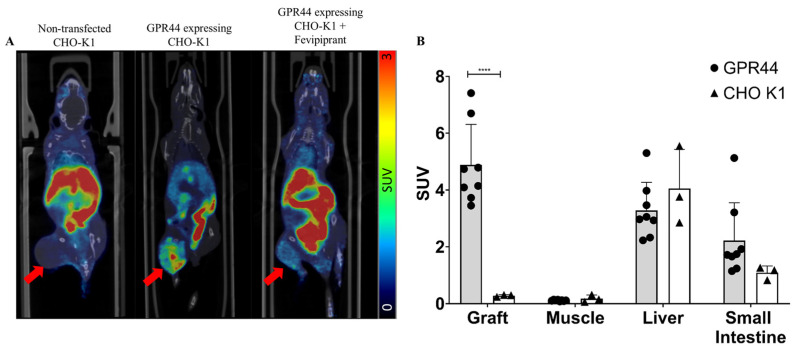
(**A**) Coronal view of fused PET/CT scan with the averaged signal of [^18^F]MK-7246 from a static scan between 60–90 min post-injection. Red arrow indicates the site of the developed tumor. (**B**) Endpoint biodistribution of [^18^F]MK-7246 in mice with tumors induced from non-transfected CHO-K1 (n = 3) or CHO-K1-expressing GPR44 (n = 8). Uptake of [^18^F]MK-7246 was significantly higher in the tumors developed from CHO-K1-expressing GPR44 (**** *p* < 0.0001).

**Figure 10 pharmaceutics-15-00499-f010:**
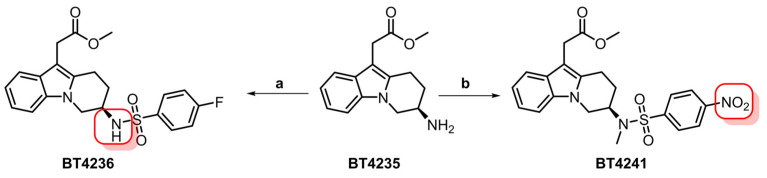
Synthesis of [^11^C]MK-7246 (BT4236) and [^18^F]MK-7246 (BT4241) precursors from the common intermediate BT4235. Red squares indicate sites for isotope incorporation. (**a**) 4-fluorobenzenesulfonyl chloride (1.9 equiv.), Et3N (3.1 equiv.), THF, rt, 5 h, 88%. (**b**) (i) 4-nitrobenzenesulfonyl chloride (2.0 equiv.), Et3N (3.1 equiv.), THF, rt, 12 h, 62. (ii) MeI (1.2 equiv.), NaHMDS (1.5 equiv.), DMF/THF, −40 °C, 3 h, 61%.

**Table 1 pharmaceutics-15-00499-t001:** Summary of [^18^F]MK-7246 production.

Entry	Precursor (mg)	A_product_ (MBq)	RCY	RP
1	0.5	345	3%	87%
2	0.5	343	3%	97%
3	0.5	332	3%	99%
4	1.0	695	7%	97%
5	1.0	571	6%	99%
6	1.0	484	5%	99%

## Data Availability

The datasets generated and analyzed during the current study are available from the corresponding author on reasonable request.

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
