# Peer review of "[18F]MK-7246 for Positron Emission Tomography Imaging of the Beta-Cell Surface Marker GPR44"

_pharmaceutics, 2023, doi:10.3390/pharmaceutics15020499_

Round 1
Reviewer 1 Report
Cheung et al. reported a successful synthesis of [18F]MK-7246, which is targeting GPR44, and its biodistribution studies and PET/MRI images on rats. Overall, the experiments were scientifically structured and the manuscript is well written. However, several concerns should be raised before publication in the journal. Especially, the authors should carefully differentiate the prove binding ability and/or specificity to GPR44-positive cells (such as pancreatic β cells) from capability to evaluate pancreatic β-cell mass.
Major comments
1. Abstract. The authors should carefully differentiate the probe binding ability and/or specificity to GPR44-positive cells such as pancreatic β cells from capability to evaluate pancreatic β-cell mass (BCM). No data suggesting the utility of [18F]MK-7246 for BCM imaging or evaluation was shown in the manuscript.
2. Introduction. Please add the detailed scientific descriptions to explain how different the GPR44 protein expression levels in the pancreas are between human and rodents (ex. ×100? ×1000?). Moreover, please discuss how the difference can affect the uptakes and imaging of the probe in the comparisons with the specificity and binding ability of the probe.
3. Figure 5B. Although higher purities of the islets seemed to show the tendency of higher probe binding, were there any significant differences among the three samples? In addition, only 98% purified islet sample showed significant difference compared with exocrine fraction, which suggested the potential limitation of the probe for evaluation of BCM and islets within the intact pancreas. Please explain this in Discussion if the authors mention the issue of BCM in this manuscript.
4. Figure 5. Please add the data regarding the GPR44 expression levels in each sample (cell line, tissue endocrine islets, and tissue exocrine fraction). Otherwise, adequate understanding of the comparisons between cell lines and tissue fraction samples was difficult.
5. In vivo PET/MRI and biodistribution (Figure 6). N=2 for PET/MRI and biodistribution study is too small, which makes the reasonable quantitative analyses in a biodistribution study difficult. Please increase the sample number for the statistical analysis.
6. In vivo PET/MRI and biodistribution (Figure 6). In PET/MRI images and biodistribution data, liver uptakes appeared even at 5 min and were stable over 150min, while small intestine uptakes were observed apparently at 30min. Although the authors described the liver uptakes were followed by excretion starting at 60 min (Line 404), is it correct? Why not 30 min or earlier?
7. Figure 7. Why did the authors determine the timing of the biodistribution as 90 min post-injection of the probe?
8. Figure 7. Although the reviewer understands the minor expression of GPR44 in rodent islets, how largely different the expression levels were in the rats which were employed in this study compared with human? Please provide the detailed information for the appropriate interpretations.
9. Figure 8. Due to the nonnegligible differences of the GPR44 (targeted molecule of the probe) expressions between rats and human, is it reasonable to calculate the human dosimetry based on rat biodistribution data for the interpretation of the probe as targeting the pancreas? Please describe the limitation on the interpretation of the dosimetry data.
10. Figure 9. Please provide the data of the GPR44 expression levels of cells employed in this study and compare with human islets or insulinoma cells.
11. The high uptake of the liver and small intestine may disturb the pancreas evaluation in vivo. Please discuss the issue.
Minor comments
1. In vitro autoradiography & Figure 5. Did the authors use human tissue for preparing pancreatic endocrine/exocrine fractions? Please specify the use of “human” samples in the appropriate parts of the manuscript and figure legend to clarify the specificity in human cells.
2. Based on the presented data, the reviewer feels that the probe should be suitable for insulinoma rather than BCM evaluation. If so, please rewrite the introduction.
3. Please discuss or introduce the references regarding 18F labeling benefits and the probe development in pancreas beta-cell imaging and/or in noninvasive BCM evaluation.
Author Response
Cheung et al. reported a successful synthesis of [18F]MK-7246, which is targeting GPR44, and its biodistribution studies and PET/MRI images on rats. Overall, the experiments were scientifically structured and the manuscript is well written. However, several concerns should be raised before publication in the journal. Especially, the authors should carefully differentiate the probe binding ability and/or specificity to GPR44-positive cells (such as pancreatic β cells) from capability to evaluate pancreatic β-cell mass.
Major comments
- Abstract. The authors should carefully differentiate the probe binding ability and/or specificity to GPR44-positive cells such as pancreatic β cells from capability to evaluate pancreatic β-cell mass (BCM). No data suggesting the utility of [18F]MK-7246 for BCM imaging or evaluation was shown in the manuscript.
The comment has been addressed and the authors made the appropriate modifications in the manuscript stating the potential of MK724 6 as a radioactive probe for GPR44 targeted PET imaging instead of pancreatic β-cell mass evaluation.
- Introduction. Please add the detailed scientific descriptions to explain how different the GPR44 protein expression levels in the pancreas are between human and rodents (ex. ×100? ×1000?). Moreover, please discuss how the difference can affect the uptakes and imaging of the probe in the comparisons with the specificity and binding ability of the probe.
In previous studies, GPR44 expression was shown to be significantly lower (up to 40 times lower) in the rat insulinoma cell line INS-1 compared with human endocrine pancreatic tissue (Hellström-Lindahl, E., Danielsson, A., Ponten, F. et al. GPR44 is a pancreatic protein restricted to the human beta cell. Acta Diabetol 53, 413–421 (2016). https://doi.org/10.1007/s00592-015-0811-3).
A lower receptor availability means a lower proportion of bound radioactive tracer to the target biomarker and thus a reduced uptake signal within the tissue of interest. Therefore, the difference in the receptor availability in human and rat pancreas further underlines the difficulties in the study of GPR44 targeted imaging of beta cells as it would require new models such as stem-cell derived islets or large animal models.
- Figure 5B. Although higher purities of the islets seemed to show the tendency of higher probe binding, were there any significant differences among the three samples? In addition, only 98% purified islet sample showed a significant difference compared with the exocrine fraction, which suggested the potential limitation of the probe for evaluation of BCM and islets within the intact pancreas. Please explain this in Discussion if the authors mention the issue of BCM in this manuscript.
Using the one-way ANOVA function with the Tukey correction for multiple comparisons (Graphpad Prism version 8.4.3), the binding level of MK-7246 to the 98% purity islets is significantly higher to 27% purity. On the other hand, the binding signal to the 76% purity islets is neither significant compared with the 98% nor the 27% purities, suggesting that MK-7246 is perhaps not sensitive enough.
However, the binding capability of MK7246 to only high-purity islets should not be a limiting factor as beta cells in the pancreatic volume are naturally dispersed. Imaging individual pancreatic islets would be impossible, but the uptake signal would be represented as the sum of all pancreatic beta cells in proportion to the whole imaged pancreas using a similarly perfused organ as reference such as the spleen. Therefore, the bottleneck would be to obtain a probe displaying a signal many folds higher in endocrine islets compared to the exocrine tissues, but also significantly higher than the non-specific signal arising from the surrounding tissues as well as the unbound plasmatic tracer associated with organ perfusion.
- Figure 5. Please add the data regarding the GPR44 expression levels in each sample (cell line, tissue endocrine islets, and tissue exocrine fraction). Otherwise, adequate understanding of the comparisons between cell lines and tissue fraction samples was difficult.
Additional graphs showing the data as relative expression ratios have been added to improve clarity.
- In vivo PET/MRI and biodistribution (Figure 6). N=2 for PET/MRI and biodistribution study is too small, which makes the reasonable quantitative analyses in a biodistribution study difficult. Please increase the sample number for the statistical analysis.
The rats' PET imaging were performed mainly to identify the optimal imaging time point for future studies in other in vivo models (i.e large animals like pigs or non-human primates) as well as to provide an overview of the tracer biodistribution. As opposed to humans, the beta cells from rats do not express high levels of GPR44, thus strong pancreatic binding was not expected. Therefore, the rats were mainly used for an overview of the clearance pathway. Additionally, more rats (n = 6) were used in the endpoint biodistribution 90 min post-injection and showed a biodistribution profile matching the one observed from the PET/MRI. Given the 3R (Reduce, Replace, Refine) principle in animal experimentation and the descriptive nature of the data, we do not deem it necessary to increase the sample number.
- In vivo PET/MRI and biodistribution (Figure 6). In PET/MRI images and biodistribution data, liver uptakes appeared even at 5 min and were stable over 150min, while small intestine uptakes were observed apparently at 30min. Although the authors described the liver uptakes were followed by excretion starting at 60 min (Line 404), is it correct? Why not 30 min or earlier?
Previous work using the isotopologue carbon-11 labeled MK-7246 and evaluation in pigs (Cheung P, Zhang B, Puuvuori E, Estrada S, Amin MA, Ye S, Korsgren O, Odell LR, Eriksson J, Eriksson O. PET Imaging of GPR44 by Antagonist [11C]MK-7246 in Pigs. Biomedicines. 2021; 9(4):434. https://doi.org/10.3390/biomedicines9040434) suggested that the excretion profile was mainly via hepatic clearance, more specifically through the gall bladder. As the decreasing signal from the liver coincides with the increased signal within the small intestine, we considered that timeframe as the active hepatic clearance of the radioactive probe through the bile. Therefore, we have described the starting point of the liver excretion of the radioactive probe at 60 min p.i because the SUV peaked at that time point before dropping, in parallel to an increase of radioactive signal within the small intestines suggesting that the MK-7246 contained in the bile reached the small intestines.
- Figure 7. Why did the authors determine the timing of the biodistribution as 90 min post-injection of the probe?
Tissue clearance of an imaging probe is a critical factor in obtaining adequate tissue-to-noise ratio and optimal imaging contrast. Therefore, we considered 90 min p.i to be optimal as the hepatic clearance of MK-7246 through the bile seemed to peak at that time point.
- Figure 7. Although the reviewer understands the minor expression of GPR44 in rodent islets, how largely different the expression levels were in the rats which were employed in this study compared with human? Please provide the detailed information for the appropriate interpretations.
Species differences in the expression of GPR44 within the pancreas have been addressed in point 2.
Low to nonexistent GPR44 expression in rats has been further shown in the present studies from the endpoint biodistribution experiment highlighting marginal pancreatic uptake of MK-7246 (SUV < 1), similar to a negative control tissue such as the muscle. Furthermore, no significant difference in the pancreatic uptake signal of MK-7246 could be observed in the presence or absence of the GPR44 antagonist Fevipiprant.
- Figure 8. Due to the nonnegligible differences of the GPR44 (targeted molecule of the probe) expressions between rats and human, is it reasonable to calculate the human dosimetry based on rat biodistribution data for the interpretation of the probe as targeting the pancreas? Please describe the limitation on the interpretation of the dosimetry data.
Despite the lack of receptor affinity in the rat’s pancreas, the main factors influencing the dosimetry come from excretion and biodistribution, so we believe the data would still be relevant. Lack of GPR44 expression within the rat pancreas would cause an underestimation of the pancreatic dose but should not significantly influence the dosimetry of the other tissues.
- Figure 9. Please provide the data of the GPR44 expression levels of cells employed in this study and compare with human islets or insulinoma cells.
The Estimated Bmax for the purchased GPR44 expressing cells is 15.0 ± 8.8 pmol/mg according to the manufacturer’s product sheet. Therefore, its expression is many folds (~1000) higher compared with the expression levels of human pancreatic endocrine tissue and INS-1 cells as reported previously by Hellström-Lindahl et al (40 and 0.88 fmol/mg respectively) referenced in point 2.
- The high uptake of the liver and small intestine may disturb the pancreas evaluation in vivo. Please discuss the issue.
We agree that the close anatomical proximity of the liver and intestines with the pancreas could potentially pose a spill-in issue, thus highly impairing the possibility of native beta-cell imaging. However, the distance between the pancreas and the liver or kidneys will be significantly larger in humans. Therefore only parts in very close proximity may be subject to spill-in. Using gallium-68 labeled Exendin4 as an example with a large spill-in from the kidneys, the issue is prevalent in smaller animals such as rats and non-human primates (5 kg), but much less significant in larger animals such as pigs (30kg) or humans (80 kg). Future studies with a GMP-compliant production method of MK-7246 and clinical evaluations would further assess those risks.
Minor comments
- In vitro autoradiography & Figure 5. Did the authors use human tissue for preparing pancreatic endocrine/exocrine fractions? Please specify the use of “human” samples in the appropriate parts of the manuscript and figure legend to clarify the specificity in human cells.
The appropriate modifications have been added to the manuscript as requested
- Based on the presented data, the reviewer feels that the probe should be suitable for insulinoma rather than BCM evaluation. If so, please rewrite the introduction.
It is reasonable to expect a possible use in this field but currently, no data is available on the GPR44 expression in insulinoma cells. (GPR44 as a Target for Imaging Pancreatic Beta-Cell Mass O.Eriksson doi: 10.1007/s11892-019-1164-z). The value of GPR44 in insulinoma is currently under investigation but unpublished pilot data indicate that GPR44 expression may be decreased in human insulinoma according to immunohistochemistry staining on biopsies.
- Please discuss or introduce the references regarding 18F labeling benefits and the probe development in pancreas beta-cell imaging and/or in noninvasive BCM evaluation.
Fluorine-18 labeled MK-7246 presents several potential advantages for in vivo PET imaging of GPR44, such as more remaining counts in the tissue due to the longer half-life of fluorine-18 (109min). Counts above background will likely be especially important for targets with low density in tissue, e.g. dispersed beta cells in the pancreas. Additionally, the Fluorine-18 nuclide has intrinsically high spatial resolution due to the lower energy and hence shorter positron range. This may be a crucial advantage in the pancreas, which is an oblong organ with a diameter of only a few centimeters. Given that the resolution of modern clinical PET scanners is in the range of 4-5 mm, the optimal resolution will minimize spillover and partial volume effects that could influence quantification.
Reviewer 2 Report
Cheung et al. describe feasibility of [18F]MK-7246-based Positron Emission Tomography imaging of the beta-cell surface marker GPR44. This study is significant from the perspective of diabetes-related research. The authors thoroughly presented the study. However, major concern arises around low number of animals used in their experiments. It is necessary to expand both rat and mouse-based studies to ensure high level of scientific rigor. Also, please address the following minor comments.
1. Abstract- Lines 20-21- You introduce MK7246 as an agent with high affinity to GPR44 before even elaborating what MK7246 is- this is the next sentence- please change the order to make it more logical
2. Introduction- Line 34- "secretion or action pathway" is not fully correct and clear; in majority of cases diabetes (persistent hyperglycemia) is a consequence of impaired insulin production due to the loss of functional beta cells (T1D) or insulin resistance in the peripheral tissues- please correct
3. Materials and Methods- Line 235 and further- please specify dosage of sevofluran used
4. Figures: Please specify numbers of experiments for each figure
5. Figures: Please present bars with dots for each individual experimen
6. Discussion- Please discuss study limitations
Author Response
Comments and Suggestions for Authors
Cheung et al. describe feasibility of [18F]MK-7246-based Positron Emission Tomography imaging of the beta-cell surface marker GPR44. This study is significant from the perspective of diabetes-related research. The authors thoroughly presented the study. However, major concern arises around low number of animals used in their experiments. It is necessary to expand both rat and mouse-based studies to ensure high level of scientific rigor. Also, please address the following minor comments.
N = 11 mice have been used in the grafted tumor model for the endpoint biodistribution experiment. Given the high effect size of 10 fold increased uptake signal of MK7246 in GPR44 expressing grafted CHO-K1 cells (SUV = ~5) compared with negative control CHO-K1 cells (SUV = ~ 0.3), we do not think a higher number of animals would be necessary to improve the scientific rigor according to the 3R (Reduce, Replace, Refine) principle in animal experimentation.
We have used n = 6 rats for the endpoint biodistribution, which has been deemed a reasonable number for providing an overall biodistribution profile for MK7246. Furthermore, given the marginal pancreatic uptake signal of MK-7246 in the pancreas, the organ of interest (SUV < 1) similar to a negative control tissue such as the muscle, in addition to an absence of blocking effect using the GPR44 antagonist Fevipiprant, we do not expect more insightful results by increasing the number of animals.
The number of animals used for imaging was indeed low (n=2 rats and n=3 mice) but were primarily used for an overview of the clearance pathway and general biodistribution in support of the endpoint biodistribution results. Given the 3R (Reduce, Replace, Refine) principle in animal experimentation and the descriptive nature of the imaging data, we have not deemed it necessary to increase the sample number in the present study.
Previous papers from the literature using similar animal amounts can be found as well:
Biodistribution in n = 5 rats https://doi.org/10.1016/j.nucmedbio.2019.04.002
Xenograft model using n = 4 mice/group https://doi.org/10.1007/s00259-022-05884-9
- Abstract- Lines 20-21- You introduce MK7246 as an agent with high affinity to GPR44 before even elaborating what MK7246 is- this is the next sentence- please change the order to make it more logical
We have performed the appropriate modification within the manuscript
- Introduction- Line 34- "secretion or action pathway" is not fully correct and clear; in majority of cases diabetes (persistent hyperglycemia) is a consequence of impaired insulin production due to the loss of functional beta cells (T1D) or insulin resistance in the peripheral tissues- please correct
We have performed the appropriate modification within the manuscript
- Materials and Methods- Line 235 and further- please specify dosage of sevofluran used
We have performed the appropriate modification within the manuscript
- Figures: Please specify numbers of experiments for each figure
We have performed the appropriate modification within the manuscript
We have performed the appropriate modification within the manuscript
- Figures: Please present bars with dots for each individual experimen
We have performed the appropriate modification within the manuscript
We have performed the appropriate modification within the manuscript
- Discussion- Please discuss study limitations
We have shown the GPR44 binding capability of MK7246 to human pancreatic islets, albeit only to the high-purity ones. It should not be an issue as beta cells in the pancreatic volume are naturally dispersed. Imaging the pancreatic islets using PET still faces many challenges, notably the small size of the islets (~20–500 µm), but the signal from the BCM would be represented as the sum of all pancreatic beta cells proportional to the whole imaged pancreas using a similarly perfused organ as reference such as the spleen. Therefore, the bottleneck would be to obtain a probe displaying a signal many folds higher in endocrine islets compared to exocrine tissues, but also significantly higher than the non-specific signal arising from the surrounding tissues as well as the unbound plasmatic tracer associated with organ perfusion. The lack of GPR44 expression in rodents’ islets limits the choice of a suitable animal model and underlines the importance of utilizing humanized animal models, large animal models, or human-derived pancreatic stem cells in the preclinical evaluation of imaging agents in diabetic research.
Round 2
Reviewer 1 Report
The authors have addressed my comments.
Author Response
We thank the reviewer for the time and consideration spent on reviewing the manuscript as well as the insightful and relevant comments.
Reviewer 2 Report
The authors addressed the comments and improved the quality of their manuscript, yet, the number of animals used in some figures- i.e., Fig.6 and 8 is still not sufficient. I am leaving the final decision with the Editor.
Author Response

(The authors gave the same response as above.)
